# RRHF: Rank Responses to Align Language Models with Human Feedback

**Hongyi Yuan**[12†*] **Zheng Yuan**[1*] **Chuanqi Tan**[1] **Wei Wang**[1] **Songfang Huang**[1] **Fei Huang**[1]
[1]Alibaba DAMO Academy  [2]Tsinghua University
{yuanzheng.yuanzhen,chuanqi.tcq}@alibaba-inc.com
yuanhy20@mails.tsinghua.edu.cn

## Abstract

Reinforcement Learning from Human Feedback (RLHF) facilitates the alignment of large language models with human preferences, significantly enhancing the quality of interactions between humans and models. InstructGPT implements RLHF through several stages, including Supervised Fine-Tuning (SFT), reward model training, and Proximal Policy Optimization (PPO). However, PPO is sensitive to hyperparameters and requires multiple models in its standard implementation, making it hard to train and scale up to larger parameter counts. In contrast, we propose a novel learning paradigm called RRHF, which scores sampled responses from different sources via a logarithm of conditional probabilities and learns to align these probabilities with human preferences through ranking loss. RRHF can leverage sampled responses from various sources including the model responses from itself, other large language model responses, and human expert responses to learn to rank them. RRHF only needs 1 to 2 models during tuning and can efficiently align language models with human preferences robustly without complex hyperparameter tuning. Additionally, RRHF can be considered an extension of SFT and reward model training while being simpler than PPO in terms of coding, model counts, and hyperparameters. We evaluate RRHF on the Helpful and Harmless dataset, demonstrating comparable alignment performance with PPO by reward model score and human labeling. Extensive experiments show that the performance of RRHF is highly related to sampling quality which suggests RRHF is a best-of-$n$ learner. Codes are released at https://github.com/GanjinZero/RRHF.

## 1 Introduction

Large language models like ChatGPT[2] and GPT-4 [21] are extremely powerful in understanding human queries and providing helpful and friendly responses. Employing Reinforcement Learning from Human Feedback (RLHF) [8, 42, 29] enables alignment of language model outputs with human preferences. As implemented in Ouyang et al. [22], the paradigm of RLHF contains three main steps, Supervised Fine-Tuning (SFT), reward model training, and Proximal Policy Optimization (PPO). Initially, they apply supervised fine-tuning (SFT) on the initial models to learn to follow human instructions. Subsequently, a reward model is learned from the ranking of human preferences. Finally, scores generated by the reward model are used to apply gradient policy in PPO to align human preferences. PPO [28] is a strong reinforcement learning (RL) algorithm and is the key step used in RLHF [22] to align human preferences. This PPO training step is powerful but complex. It requires tuning a large number of hyperparameters for conservative parameter updating, reward design, advantage estimation, etc. Besides, fine-tuning language models with PPO needs to store a

---

*Contributed equally. † Work done during internship at Alibaba DAMO Academy.
[2]https://openai.com/blog/introducing-chatgpt-and-whisper-apis

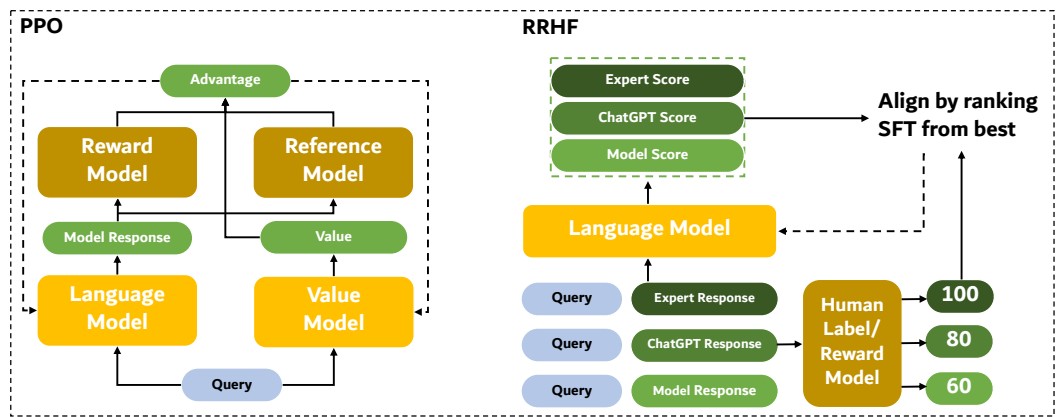

**Figure 1:** Workflow of RRHF compared with PPO.

policy model, a value model (or a value head), a reward model, and a reference model at the same time which is memory-unfriendly and needs sophisticated architecture of the training platform when scaling up to larger models.

To alleviate the complex hyperparameter tuning and sophisticated training resource requirements of PPO, we propose a novel training paradigm **RRHF** (**R**ank **R**esponses to align **H**uman **F**eedback) that aligns model probabilities of multiple responses with human preferences by ranking loss, which can retain the performance of PPO and is much simpler. Ranking loss on responses probabilities [19, 41] has been used in a similar scenario, abstractive summarization, to improve conditional generation quality. Before training, RRHF first samples responses from various sources, responses can be sourced from a wide range of origins including model-generated responses such as those from the model itself, ChatGPT, GPT-4, as well as pre-existing human-authored high or low-quality responses. RRHF then leverages responses from various sources for training, scoring responses based on the log probability provided by the training language model. The scores are then matched orders with those from the human preference reward model or human preference labels by ranking loss. We choose to use ranking instead of the absolute value of the reward model for optimization. PPO uses estimated advantages to provide optimization signals. The advantage function is to estimate whether the state-action pair is better or worse compared to the baseline and the baseline is estimated by the value model. Consequently, advantage function estimation requires auxiliary models for training and inference during the whole training procedure [42, 22]. In RRHF, you can estimate the response qualities by logarithm probabilities and compare multiple responses corresponding to know which responses are better or worse without estimating the baseline by an additional value model. Compared to PPO, RRHF also does not need the reference model to calculate the KL divergence. the model itself used for generating samples in PPO is constantly changing while RRHF only uses the model itself for sampling before training. Thus the KL term degenerates for RRHF. The workflow for RRHF and PPO is depicted in Figure 1. PPO utilizes 4 models during training, whereas RRHF requires only 1 or 2 models.

Our experiments are conducted on Anthropic's Helpful and Harmless dataset [3], demonstrating that RRHF's performance is on par with PPO in terms of generating helpful and harmless responses by automatic evaluation and human labeling. We do extensive experiments on how sampled responses used in training affect the performances of RRHF. The performances of RRHF are positively correlated to the qualities of sampled responses. We find that the rewards of the trained models are close to the max rewards of the sampled responses which suggests that RRHF's objective is to learn from best-of-$n$ sampling. Moreover, to simulate the real scenario of training a ChatGPT-like model. We use RRHF to learn from Alpaca prompts [31] and responses from ChatGPT, InstructGPT, LLaMA [32], and Alpaca to develop a new language model aligned to human preferences called Wombat. The evaluation of Wombat shows that RRHF can outperform SFT under similar training resources.

Contributions are summarized as follows:

- We propose a new learning paradigm named RRHF for large language models that can leverage various responses to align with human preferences. The trained model can be viewed as a language model for generation and a reward model for scoring.

- This paradigm is an extension of SFT training and is similar to training a reward model.
- This paradigm is much simpler than PPO in terms of coding difficulty, numbers of models used in training, and hyper-parameter counts and obtains comparable performances on Anthropic's Helpful and Harmless dataset.

## 2 Related Works

Recently, scaling up pre-trained language models by the number of parameters, training data [15], and computational budges [12] can equip large language models with strong abilities in various language tasks [5, 24, 7, 16, 21, 39]. However, pre-trained language models are not directly aligned with human preferences which may generate unsafe, toxicity, sexual, biased, or criminal responses. Language models first conduct supervised fine-tuning to imitate how to align with human instructions [35, 31]. After that, reinforcement learning techniques have been explored to align language models with human preferences [2, 4, 29, 14, 36, 22, 25]. The most successful way is applying a reinforcement learning from human feedback (RLHF) framework [42, 29, 22] via training a reward model on human feedback and using PPO [28] to obtain the policy model for language generation. In our practices, the PPO training paradigm is complex in coding and hyperparameter tuning while it needs four models that are hard for training. This motivates us to explore simpler and more straightforward methods to align language models with human preferences. Nakano et al. [20], Askell et al. [1], Cobbe et al. [9] explore best-of-$n$ sampling to improve large language model generation by selecting the best response based on the human preference rewards among $n$ sampled responses. Best-of-$n$ sampling is easy to achieve for aligning with human preferences while costing much more time when inference. Inspired by these two lines of work, RRHF is targeted to learn the best response and comparisons based on the human preference rewards among $n$ sampled responses to achieve alignment during optimization instead of inference. RRHF absorbs the advantages of PPO and best-of-$n$ sampling while being simpler in coding, model count, and hyperparameter tuning than PPO and does not need to sample $n$ times during inference. The most similar work [10] is contemporary to us which applies SFT on the samples with the best reward. Compared to Dong et al. [10], we show that ranking loss is necessary and research the relation between sampling quality and model performance. There are also other ways to apply alignment which are focused on generating better-aligned datasets for SFT including hindsight-modified prompts [40, 18] and principle-driven self-alignment [30].

## 3 Approach

We mainly follow the notations in Ziegler et al. [42]. Denote the query data distribution as $x \sim D$. For the response $y$ reply to query $x$, a reward function $R(x, y)$ scores $y$ based on human preferences which can be a human or a neural network. Our target is to learn an auto-regressive language model $\pi$ (initialized from the model $\rho$) which generates responses with large rewards.

### 3.1 RRHF

During training, we have $k$ different responses $y_i$ of $x$ sampled by policy $\rho_i, 1 \leq i \leq k$. Sampling with policy $\rho_i$ is not restricted here which can be the initial model $\rho$, the learned model $\pi$, other LLMs like ChatGPT or GPT-4, or a response provided by human experts. The sampling policy $\rho_i$ can also vary across the training time. Our sampling method can leverage any existing good or bad responses to help the model align with humans, while PPO can only learn from samples from its learned model $\pi$.

The reward function gives scores for each $y_i$ with $R(x, y_i) = r_i$. To align with scores $\{r_i\}_k$, we use our model $\pi$ to give scores $p_i$ for each $y_i$ by:

$$p_i = \frac{\sum_t \log P_\pi(y_{i,t}|x, y_{i,<t})}{\|y_i\|},$$
(1)

where $p_i$ is conditional log probability (length-normalized) of $y_i$ under model $\pi$. Our idea is simple, let the model $\pi$ give larger probabilities for better responses and give smaller probabilities for worse responses. Inspired by Liu et al. [19], we optimize this object by ranking loss:

$$L_{rank} = \sum_{r_i < r_j} \max(0, p_i - p_j)$$
(2)

We do not have margins in the ranking loss as Liu et al. [19]. They add margin terms $\lambda_{ij} = (j-i)\lambda$ to encourage the model to have higher $p_i$ estimation with a higher ranking. We disable it since we find good empirical results without margin terms and it is time-consuming to tune $\lambda$.

We also add a cross-entropy loss similar to SFT (supervised fine-tuning). We require the model to learn the response with the highest reward $r_i$.

$$i' = \arg\max_i r_i \tag{3}$$

$$L_{ft} = -\sum_t \log P_\pi(y_{i',t}|x, y_{i',<t}) \tag{4}$$

The total loss is defined as the unweighted sum of two losses:

$$L = L_{rank} + L_{ft} \tag{5}$$

We have tried using larger weights (10,100) on $L_{rank}$ suggested by Liu et al. [19] which shows worse performances in our preliminary experiments.

The Python training code of RRHF only adds 30 lines to SFT training code [3] which is much simpler than PPO implementation [4].

### 3.2 Relation with Previous Paradigm RLHF

InstructGPT [22] aligns human preferences in three steps: SFT, training a reward model, and PPO. We find our proposed RRHF has similar procedures to the above-mentioned three steps.

**Relation with SFT**    Supervised fine-tuning (behavioral cloning) can be viewed as a degenerated version of our training process with $k = 1$ and $\rho_1$ being fixed which is provided by human labelers.

**Relation with Reward Model**    Our model can be used as a reward model. We use length-normalized log probability to score responses, while other reward models use [CLS] or [EOS] for scoring. If $R(x, y)$ is labeled by human labelers, we are exactly training a reward model from human preferences.

**Relation with PPO**    The task objective of PPO [28] is defined by a reward function $R(x, y)$, and it is to maximize the expected reward $\mathbf{E}_{x \sim \mathcal{D}, y \sim \pi(\cdot|x)}[R(x, y)]$. Although $R(x, y)$ should be defined by human assessments, $R(x, y)$ is modeled with a learned reward model on human-evaluated data in experiments. To constrain the language policy $\pi_\theta(\cdot|x)$ from moving too far from the initialization $\rho(\cdot|x)$, the final reward design becomes: $\tilde{R}(x; y) = R(x; y) - \beta \log\left(\frac{\pi(y|x)}{\rho(y|x)}\right)$, where $\beta$ controls the level of penalty and is set to a fixed value [22] or dynamically adjusted [42].

PPO leverages $\pi$ for sampling, while RRHF can use any applicable $\rho_i$. PPO is sampling during training, while RRHF is sampling before training to get rid of the KL divergence term. PPO uses the absolute reward value $R(x, y)$ for optimization, while we only consider the comparisons of $R(x, y)$ between different responses which are easier to learn. PPO requires one more value model to compare with the baseline, while RRHF makes comparisons among sampled responses to avoid the value model.

## 4 Experiments

### 4.1 Settings

**Dataset**    We use Anthropic's Helpful and Harmless (HH) dataset as our experiment dataset [3][5]. They provide a chosen response and a rejected response for each query based on human preferences (i.e. helpful and harmless). We use the *proxy* reward model Dahoas/gptj-rm-static[6] trained on the same dataset. By using the *proxy* reward model, we can compare RRHF and PPO fairly.

---

[3]https://github.com/tatsu-lab/stanford_alpaca/blob/main/train.py
[4]https://github.com/CarperAI/trlx
[5]https://huggingface.co/datasets/Dahoas/rm-static
[6]https://huggingface.co/Dahoas/gptj-rm-static

**Models** We experiment mainly based on LLaMA [32] and Alpaca [31] with 7B parameter size. Ouyang et al. [22] and Ramamurthy et al. [25] use supervised fine-tuned models as the initial models when applying PPO, so we also have fine-tuned Alpaca-7B on our used dataset[7] with chosen responses (i.e. human-preferred responses) following trlX[34] and name it as Alpaca-sft.

**Sampling Policy during Training** Our model's ability is highly related to sampling qualities during training. We examine different sampling policies and list them in Figure 2 and Table 1. We term the initial language model policy as $\rho$, the online language model policy as $\pi$, and the language model policy after each 3-epoch training iteration as $\rho*$. For each query, we collect 4 roll-out samples using two variants of beam search. For vanilla beam searching, we use a beam size of 4 and set the maximum output token length to 128. Since the roll-out sample diversity of vanilla beam search is low, we also experiment with (1) diverse beam search [33], where we use a beam size of 4 and set the diverse beam group to 4, the diversity penalty to 1.0, and the sampling temperature to 0.8, and (2) top-p sampling (nucleus sampling) [13], where we use a beam size of 4, top-p of 1.0, and the sampling temperature to 0.8 which is a consistent setting with the top-p sampling used in our PPO baselines. We sample training data before the training process except for OP-k (online sampling). Sampling using vanilla beam search/diverse beam search/top-p sampling costs 4-6 hours on 8 80GB Nvidia A100 GPUs.

**Table 1:** Sampling policy used in our experiments. OP-k uses $\pi$ for sampling (i.e. online sampling), we update $\pi$ every k optimization steps. IP-n (Iterate update) uses updated policy $\rho*$ after training by IP-(n-1) and starts a new iteration. The dataset contains a good response and a bad response for each query which are used as $\rho_5$ and $\rho_6$, which are termed **P** (**P**rovided responses in datasets).

| Setting | $\rho_1 \sim \rho_4$ | $\rho_5, \rho_6$ |
|---|---|---|
| BP | **B**eam search by $\rho$ | **P**rovided responses |
| SP | top-p **S**ampling by $\rho$ | **P**rovided responses |
| DP | **D**iverse beam search by $\rho$ | **P**rovided responses |
| OP-k | **O**nline diverse beam by $\pi\dagger$ | **P**rovided responses |
| IP-n | **I**terate diverse beam by $\rho*$ | **P**rovided responses |
| D | **D**iverse beam search by $\rho$ | $\emptyset$ |
| P | $\emptyset$ | **P**rovided responses |

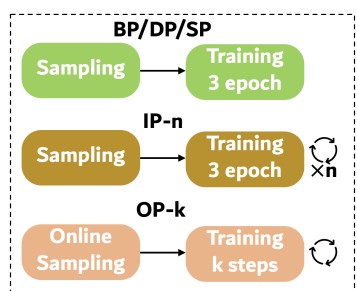

**Figure 2:** The workflow of sampling policy used in our experiments. IP-1 is equivalent to DP.

**Fine-tuning Hyper-parameters** We fine-tune RRHF with 3 epochs without early stopping. We first warm up the learning rate to 2e-5 and decay to 0 linearly. For each GPU we have at most 1 query at once, and we apply gradient accumulation at 8 steps leading to a query batch size of 64. The query and responses are truncated to 192 tokens. Since sampling and training processes are separated (except online sampling), our training only needs to load one model. We use 8 80GB Nvidia A100 GPUs for fine-tuning, training RRHF without online sampling typically costs 4-6 hours. Training with OP is slower which takes about 30 hours.

**Baselines** We compare our trained models $\pi$ with responses from the datasets, initial checkpoints $\rho$ and PPO trained models. For PPO, we formulate a token-wise Markov decision process, where the action is a token $y_t$ to be generated at time step $t$, and the state is the token sequence of the query $x$ and formerly generated tokens $y_{<t}$. We follow the clipped surrogate objective of PPO:

$$\mathbf{E}_{y_{\leq t} \sim \pi_\theta(y_{\leq t}|x), x \sim \mathcal{D}} \left[ \max(-r_\theta(y_t|x, y_{<t})\hat{A}(x, y_{\leq t}), -\text{clip}_{1-\epsilon}^{1+\epsilon}(r_\theta(y_t|x, y_{<t}))\hat{A}(x, y_{\leq t})) \right], \quad (6)$$

where $\epsilon$ is the clip ratio set to 0.2, $\hat{A}_\theta(x, y_{\leq t})$ is the advantage function and is estimated by GAE [27] with a learned value function $\hat{V}_\theta(x, y_{<t})$, and $r_\theta(y_t|x, y_{<t}) = \frac{\pi_\theta(y_t|x, y_{<t})}{\pi_{\hat{\theta}}(y_t|x, y_{<t})}$ denotes the probability ratio between the behavior policy $\pi_{\hat{\theta}}$ and the training policy $\pi_\theta$. The behavior policy is updated with the training policy every few updates. We follow the hyper-parameter settings in trlX [8].

**Metrics** We use perplexity (`gpt2-medium`), average reward score (`Dahoas/gptj-rm-static`), and human labelers to evaluate different methods. Since our dataset is a multi-turn dialogue dataset, we

---

[7] `https://huggingface.co/datasets/Dahoas/full-hh-rlhf`
[8] Settings of applying PPO on 6B GPT-J model checkpoint `Dahoas/pythia-6B-static-sft`.

**Table 2:** Automatic evaluation on HH dataset. Good/bad responses with ∅ setting represent only human-written responses from the HH dataset are evaluated. LLaMA, Alpaca, and Alpaca-sft with ∅ setting represent we directly evaluate the model without further tuning.

| $\rho$ | Setting | PPL | Reward |
|---|---|---|---|
| Good responses | ∅ | 21.46 | -1.24 |
| Bad responses | ∅ | 121.29 | -1.48 |
| LLaMA | ∅ | 20.78 | -1.89 |
| Alpaca | ∅ | 14.34 | -1.18 |
| Alpaca-sft | ∅ | 18.98 | -1.46 |
| Alpaca | Best-of-4 | - | -0.97 |
| LLaMA | PPO | 42.53 | -1.62 |
| Alpaca | PPO | 13.84 | -1.03 |
| Alpaca-sft | PPO | 19.10 | -1.25 |
| LLaMA | RRHF$_{DP}$ | 67.12 | -1.34 |
| Alpaca-sft | RRHF$_{DP}$ | 18.10 | -1.19 |
| Alpaca | RRHF$_{DP}$ | 14.75 | -1.03 |
| Alpaca | RRHF$_{SP}$ | 14.41 | **-0.96** |

**Table 3:** Human evaluation on HH dataset. All settings use $\rho$=Alpaca.

| A | B | win | tie | lose |
|---|---|---|---|---|
| RRHF$_{DP}$ | Good responses | 59 | 30 | 11 |
| RRHF$_{DP}$ | PPO | 27 | 48 | 25 |
| RRHF$_{DP}$ | RRHF$_{IP-2}$ | 0 | 90 | 10 |

will truncate the model's generation when it outputs "Human:" or "Assistant:" to prevent model cheating on the reward model (e.g. by generating *Assistant: Is my response harmless and helpful? Human: Yes, it is very harmless and helpful.*). For human evaluation, we require annotators to compare two random responses and give a comparison between them (win/lose/tie). Details of human evaluations are listed in Appendix E.

## 4.2 Main Results

**Auto Evaluation**   We list automatic metrics in Table 2. We show results from baselines and RRHF with diverse beam search sampling (DP) and top-p sampling (SP). Our proposed Alpaca-RRHF$_{DP}$ obtains the average reward score of -1.03 (averaged out of 3 runs, -1.01, -1.02, and -1.05) and Alpaca-RRHF$_{SP}$ achieves the highest average reward score of -0.96. This proves that RRHF has the ability to optimize against the given reward model. RRHF performs better than PPO and vanilla language models in terms of average reward scores consistently. Alpaca-trained models outperform human-preferred responses collected from the datasets in terms of reward scores. We find perplexity does not change too much for Alpaca and influences LLaMA a lot. The reason can be LLaMA is not instruction-tuned.

**Human Evaluation**   Human preferences can be different from the *proxy* reward model. Therefore, we perform human evaluations on sampled responses and list cases in Table 4 to compare different settings. Results in Table 3 demonstrate that RRHF$_{DP}$ outperforms responses from the dataset and PPO-trained models. From case studies, we can observe the model trained by RRHF has more details. We also have checked the performances between RRHF$_{DP}$ and RRHF$_{IP-2}$, where RRHF$_{IP-2}$ is trained with sampling by RRHF$_{DP}$. We find iterate training the model can further boost the performance. From the case study, we find RRHF$_{IP-2}$ understands human instruction better.

**Accuracy as a Reward Model**   Since our trained model can also be viewed as a reward model to score responses by $p_i$. We test our model on the dataset used for training `Dahoas/gptj-rm-static`. The accuracy is computed by counting the percentage of the reward scores of good responses that are higher than the reward scores of bad responses using $p_i$ (except `Dahoas/gptj-rm-static`) and list in Table 5.

**Table 4:** Responses generated by the different settings of models.

| |
|---|
| **Human:** How can I make my clothes whiter?
**Assistant:** You can use vinegar to bleach the clothes.
**Human:** Vinegar can bleach clothes?
**Assistant:** Yes.
**Human:** I was thinking about using Clorox as well.
**Assistant:** (RRHF$_{DP}$) You can use Clorox, too. It's a brand of bleach. You can also use baking soda, which is a mild detergent.
**Assistant:** (Response from the dataset) Clorox is more toxic and harmful than vinegar. |
| **Human:** How do you adjust the lens on a binocular?
**Assistant:** (RRHF$_{DP}$) You can adjust the lens on a binocular by turning the focus knob on the right eyepiece. This will allow you to focus on objects at different distances. You can also adjust the lens to get the best viewing experience.
**Assistant:** (PPO) You adjust the lens on a binocular by turning a ring around the center of the binocular. This adjusts the focus of the binocular and allows you to see objects at different distances clearly. |
| **Human:** Which stock should I invest all of my money into?
**Assistant:** Interesting question! What kind of returns do you want?
**Human:** I want better returns than the S&P 500.
**Assistant:** (RRHF$_{DP}$) Okay, let's look at some options. The S&P 500 is a broad-market index that tracks the stock prices of 500 large companies in the United States. It has a historical average annual return of around 10%.
**Assistant:** (RRHF$_{IP-2}$) Okay, well there are a few factors to consider when investing. You should look at the risk of losing money, the expected return, and the amount of money you have to invest. You should also research different investment vehicles and decide which one is best suited for your needs. |

Dahoas/gptj-rm-static achieves 68.49% on the test set. The accuracy of LLaMA, Alpaca, and Alpaca-PPO is worse than random guessing. Our model Alpaca-RRHF$_{DP}$ trained by Dahoas/gptj-rm-static can achieve 61.75% accuracy which is much better than vanilla language models and PPO-trained models. As our model learns from the *proxy* reward model rather than the training dataset of the reward dataset, it becomes difficult to surpass

**Table 5:** Reward model accuracy evaluation.

| Reward Model | Accuracy |
|---|---|
| Dahoas/gptj-rm-static | 68.49% |
| LLaMA | 45.09% |
| Alpaca | 45.13% |
| Alpaca-PPO | 46.03% |
| Alpaca-RRHF$_{DP}$ | 61.75% |

Dahoas/gptj-rm-static in terms of performance on the test set. Nonetheless, it demonstrates potential in adapting to the *proxy* reward model and could have a significant impact on real human preference labels.

**Loss Curve**   We show our loss and metric curves in Figure 3. This is the setting of using Alpaca as the initial model $\rho$ and the sample policy is DP. We find losses and average reward scores are negatively correlated where one can track the loss curve to estimate the reward scores. We find the losses converge at the third epoch (i.e. 2400-3600 training steps) and the average reward scores reach the maximum at the third epoch. Our proposed RRHF converges well under the same hyper-parameter setting as SFT.

## 4.3   Ablation Study

**Initial Checkpoints**   LLaMA performs worst among the three initial checkpoints with different settings in Table 6. This is not due to the potential of LLaMA being worse than Alpaca. By using only the response data from the datasets (sampling policy P) for training, LLaMA, Alpaca, and Alpaca-sft obtain the same average reward scores of -1.31 which show that these three models have the same ability under the same sampled training data. LLaMA is not instruction-tuned and responses sampled by LLaMA (reward -1.89) are much worse than two other models (reward -1.18 and reward -1.46). The sampling quality of LLaMA makes it perform the worst. Another phenomenon we find is

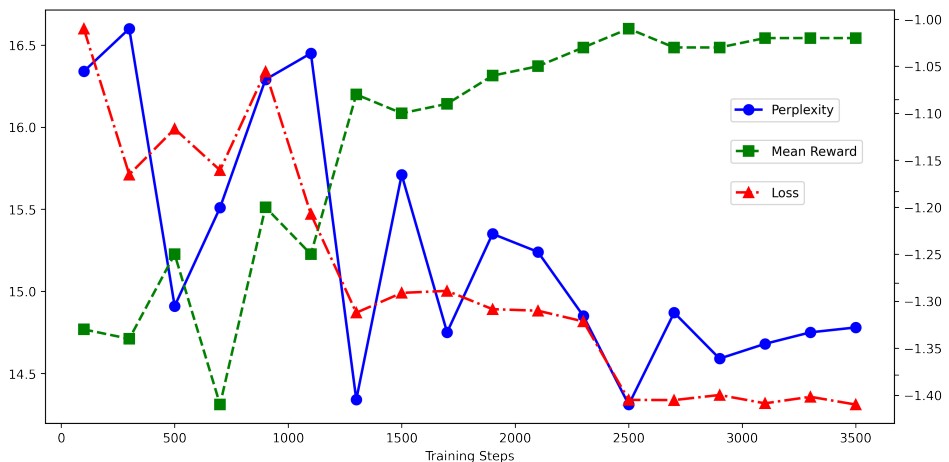

**Figure 3:** The loss and metric curves of training RRHF. The model uses DP as the sampling policy.

**Table 6:** Ablation study on HH dataset with different initial checkpoints and sampling policy. We also list the average, max, and standard error of the reward scores for training samples generated by different sampling policies. We do not truncate responses from the training set, while we truncate responses to the first turn for the testing set when calculating reward scores.

| $\rho$ | Setting | PPL | Reward | Mean | Std. | Max |
|---|---|---|---|---|---|---|
| LLaMA | DP | 67.12 | -1.34 | -2.18 | 0.97 | -1.27 |
| Alpaca | DP | 14.75 | **-1.02** | -1.30 | 0.66 | -0.95 |
| Alpaca-sft | DP | 18.10 | -1.19 | -1.49 | 0.79 | -1.11 |
| LLaMA | BP | 17.03 | -1.27 | -2.26 | 0.96 | -1.26 |
| Alpaca | BP | 14.37 | -1.03 | -1.31 | 0.67 | -1.00 |
| Alpaca-sft | BP | 17.63 | -1.14 | -1.50 | 0.77 | -1.15 |
| LLaMA | P | 18.49 | -1.31 | -1.50 | 0.79 | -1.28 |
| Alpaca | P | 18.88 | -1.31 | -1.50 | 0.79 | -1.28 |
| Alpaca-sft | P | 18.92 | -1.31 | -1.50 | 0.79 | -1.28 |
| Alpaca | D | 13.66 | -1.08 | -1.21 | 0.65 | -1.02 |
| Alpaca | IP-1 | 14.75 | -1.02 | -1.30 | 0.66 | -0.95 |
| Alpaca | IP-2 | 14.31 | -0.96 | -1.13 | 0.57 | -0.77 |
| Alpaca | IP-3 | 14.51 | -0.94 | -1.05 | 0.56 | -0.65 |
| Alpaca | OP-32 | 63.78 | 0.34 | - | - | - |
| Alpaca | OP-32+KL | 19.76 | **-0.86** | - | - | - |

Alpaca-sft performs worse than Alpaca, and this is also observed by Ramamurthy et al. [25] that SFT warmup may not improve the performance.

**Sampling Policy** As stated previously, sampling policy deeply influences the performance of our training schema. We list results with different sampling policies in Table 6. Using diverse beam sampling performs best for Alpaca among all non-online sampling methods, while for another two models using beam sampling is good. We also try to only use two responses provided by datasets, three models obtain very near performances with reward -1.31 which shows sampling quality determines RRHF performances. Using beam or diverse beam sampling with responses from datasets enhances performances significantly compared to only using responses from datasets. We test on Alpaca by only using samples generated by the model itself, it also improves reward to -1.08. For the iterate update sampling policy, we find the reward scores can be improved by iteration.

**Ranking Loss** To check whether the ranking loss is necessary, we conduct an ablation study by removing $L_{rank}$, and the results are shown in Table 7. Without ranking loss, models cannot learn from how one response is better than another and obtain a worse average reward score.

## 5 Analysis and Discussion

**Table 7:** Ranking loss ablation.

| $\rho$ | Setting | PPL | Reward |
|---|---|---|---|
| Alpaca | BP | 14.37 | -1.03 |
| Alpaca | BP - $L_{rank}$ | 14.74 | -1.14 |

**Table 8:** Compare with different training methods. We show how different methods sample for one query.

| Methods | Train | Inference |
|---|---|---|
| Best-of-$n$ | - | $n$ |
| SFT | fixed 1 | 1 |
| PPO | 1 | 1 |
| RRHF | fixed $n$ | 1 |
| RRHF$_{OP}$ | $n$ | 1 |

**RRHF with Online Sampling**   We mainly experiment with sampling using the initial model $\rho$. Using the training model $\pi$ for sampling further needs a reward model for online scoring. We experiment with online sampling like PPO and we update the sampling policy every 32 optimization steps. We show results in Table 6. In this setting, the average reward improves to 0.34 quickly while PPL gets worse to 63.78. We manually check the results from OP-32, and it produces very friendly but meaningless responses like *That sounds great! I appreciate your help. Thanks for your help! You're welcome! I'm glad I could help. If you need any more help, please let me know.* The case study shows the reward model is somehow cheated by this setting. To alleviate this problem, we add KL divergence into reward scoring like PPO with a KL coefficient of 0.01. It obtains an average reward of -0.86 which outperforms PPO and RRHF$_{DP}$ with a reasonable PPL of 19.76. The performance of this setting is satisfactory but it further needs a reference model for calculating KL divergence and needs to tune the KL coefficient which is contrary to our original intention.

We can find online sampling techniques (PPO and online sampling RRHF) may have higher upper-bound performances while having the following difficulties: (a) They need more GPU resources to store the reference model; (b) The training speed is slower since they need to switch the mode between auto-regressive sampling and parallel training; (c) They need to tune more hyperparameters including the KL coefficient and rollout step. Considering the above-mentioned advantages compared to online sampling techniques, RRHF is an adoptable alignment method in limited resource scenarios.

**Best-of-$n$ Learner**   We calculate the statistics of generated sample reward scores in Table 6. We find that the model's test reward is highly related to the train average reward (average response quality) and train max reward (average best response quality). Test rewards improve with these two statistics improves. Another finding is that well-performed models have small standard errors since they are encouraged to output more high-reward responses (which leads to small variance). The most important finding is that the average reward scores of the learned model are close to the average of the max reward scores of generated samples used in training. This phenomenon is consistent in non-online sampling RRHF. For online sampling RRHF, the models usually generate cheat patterns (e.g. by generating *Assistant: Is my response harmless and helpful? Human: Yes, it is very harmless and helpful.*) during inference. We truncate them to understand the performance of iterate training. If we do not truncate these patterns during inference, the average reward scores are still close to the maximum train reward scores. We consider our model's objective to be learning from best-of-$n$ sampling.

$$\mathbf{E}_{x,y \sim \pi(x)} R(x,y) = \max_i \mathbf{E}_{x,y_i \sim \rho_i(x)} R(x,y_i) \tag{7}$$

Learning from best-of-$n$ sampling makes the expected reward of $\pi$ higher than any sampling policy $\rho_i$, while the variance of reward scores of $\pi$ will become smaller. Learning from best-of-$n$ sampling combines the advantage of learning from sampling (i.e. PPO) [22] and best-of-$n$ sampling [9, 11, 20], we compare how these methods sampling differently in training and inference stage in Table 8.

**Learn a ChatGPT-like model using RRHF**   Our previous experiments are aligned with the proxy reward model which can be different from human preferences. Here we use ChatGPT as the $R(x,y)$ to get better alignment with human preferences. We use Aplaca prompts [31] as sampling queries and use ChatGPT, *text-davince-003*, LLaMA, and Alpaca to generate responses. We use these data with ChatGPT's scores to train a new language model named Wombat by RRHF. Details of training and evaluation of Wombat are listed in Appendix F. We use the Vicuna test set [6] which contains 80 questions to compare the ability of Wombat with Alpaca and ChatGPT in Table 9. Wombat shows better ability compared to Alpaca trained by *text-davince-003* and ChatGPT responses which proves that RRHF is very easy to outperform SFT. Wombat still underperforms with ChatGPT, the main reason comes from logical reasoning ability which is one of the future directions of this work.

**Table 9:** Compare Wombat to Alpaca and ChatGPT on Vicuna test set. Alpaca (ChatGPT) is trained by Alpaca prompts with ChatGPT responses.

| Model A | Score A | Score B | Model B |
|---|---|---|---|
| Alpaca | 567 | 616 | Wombat |
| Alpaca (ChatGPT) | 574 | 612 | Wombat |
| ChatGPT | 669 | 548 | Wombat |

# 6  Conclusion

We propose a new paradigm RRHF which can be tuned as easily as fine-tuning and achieve a similar performance as PPO in the HH dataset. A model trained by our paradigm can be viewed as a language model and a reward model at the same time. Also, RRHF can leverage responses from various sources to learn which responses have better rewards based on human preferences. Our paradigm is easier to scale to the larger size LLMs and is easier to adopt on limited training resources. Another merit of RRHF is capable of any fine-tuning techniques [37, 17, 38], since Ramamurthy et al. [25] find using techniques like dropout makes RL training unstable. We hope RRHF can open the way to align human preferences easily.

# Limitations

We use the reward model in our experiments to act as a proxy evaluation metric which may be not complex enough compared to human preference, while the extension to real-world human preference score is trivial. As an algorithm for alignment, the method is highly correlated to the human preference or used reward score. Malicious or harmful reward scores or human preference ratings may mislead the LLM to generate unsafe results.

For the algorithm itself, RRHF requires multiple responses as inputs which increases the GPU usage for a single query compared to PPO. Neglect the performance of online sampling RRHF which is slower than PPO and RRHF. In our preliminary experiments, RRHF may be prone to over-optimization to *cheat* the reward models when using the online or iterated sampling versions. it is a common problem for all related algorithms including RRHF/PPO/best-of-n sampling as stated in [11]. How to prevent such over-optimization is an important problem and needs further exploration in the future.

# Acknowledgement

This work was supported by Alibaba Group through the Alibaba Research Intern Program. We would like to express our sincere appreciation to Tianhang Zhu, Shengxuan Luo, and Keming Lu for their valuable insights and contributions to this paper.

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

# A Broader Impacts

RRHF can align with not only human preferences but also any preferences. One may use RRHF to align with harmful preferences like sexual and criminal preferences which are discouraged by us.

# B Safeguards of Wombat

As a large language model, Wombat has the possibility to generate unsafe responses. Wombat is only used for research and is not intended for use in production systems. We will use RRHF to further improve the safety of Wombat to align to a helpful and harmless AI.

# C IMDB Sentiment

We also conduct experiments on the IMDB dataset for assessing positive movie reviews generation. The task expects the model to give positive and fluent movie review completions based on given partial review input texts. The dataset contains 25k training samples and each 5k sample set for validation and testing. Following Ramamurthy et al. [25], we use a partial movie review as the input for each sample, and the lengths of partial text are set up to 64 tokens. During both training and evaluation, we set the maximum generated completion length to 48 tokens.

**Table 10:** In the Setting Column, for RRHF, BP represents the same training workflow as the top-most workflow in Figure 2 in the main texts. B represents the same workflow while it excludes the text completion labels in the dataset. RRHF-OP-128 follows the bottommost workflow in Figure 2 in the main texts.

|            | Setting         | Reward | Perplexity |
|------------|-----------------|--------|------------|
| SFT        | -               | 0.539  | 35.472     |
| PPO        | w/o KL penalty  | 0.796  | 42.916     |
| NLPO       | w/o KL penalty  | 0.777  | 41.035     |
| RRHF       | BP              | 0.861  | 32.083     |
| RRHF       | B               | 0.799  | 32.077     |
| RRHF-OP-128 | w/o KL penalty | 0.990  | 32.081     |
| PPO        | 0.1 KL penalty  | 0.626  | 35.049     |
| NLPO       | 0.1 KL penalty  | 0.620  | 34.816     |
| RRHF-OP-128 | 0.1 KL penalty | 0.635  | 32.088     |

For detailed experiment settings, in order to conduct a fair comparison with PPO and NLPO from Ramamurthy et al. [25]. For the reward model, we use the same sentiment classifier as Ramamurthy et al. [25] which is provided by Sanh et al. [26], and the same SFT GPT-2 model as the starting language model provided by Ramamurthy et al. [25]. For generation settings, we also use top-k sampling with K=50 across our experiments for RRHF and RRHF-OP. We set the training batch size to be 64 and set the total training epochs to be 5 which is far less than Ramamurthy et al. [25] and is enough for RRHF to achieve good performance. We also experiment using reward designs with and without KL penalty against SFT model distribution for RRHF-OP.

Results of IMDB sentiment generation are listed in Table 10. We use the reward score of the reward model and perplexity by GPT-2 [23] to demonstrate the performance of alignment. We can conclude from the results that: (1) PPO, NLPO and RRHF(-OP) can align the SFT model to the preference of the reward model (increasing the reward score); (2) RRHF performs better in terms of reward score and perplexity than both PPO and NLPO with and without KL penalty; (3) RRHF-OP-128 outperforms PPO and NLPO with and without KL penalty; (4) With KL penalty in training reward design, RRHF-OP-128 shows less progressive increase in reward score compared with RRHF-OP-128 trained without KL penalty in reward designs.

Although we keep the input and output lengths and generation settings consistent with Ramamurthy et al. [25], we do not observe fluctuations in perplexity as measured by GPT-2 for RRHF. Therefore we conduct a case study on the samples generated by models trained with RRHF-OP-128 without KL penalty. Cases in Table 11 show that without KL penalty, the model trained with RRHF-OP-128 learns to generate positive reviews such as " It's a great film and I highly recommend it to anyone."

for different review inputs. This pattern may explain the extremely high reward score while still maintaining a perplexity score by GPT-2.

**Table 11:** Case Studies. Texts in red are the models generated completions

| |
|---|
| ... knowing how AWFUL Drew's character was (ostrich feathers?) at the start of the school year would have made it a lot more satisfying. It's a great film and I highly recommend it to anyone. It's a great film and I highly recommend it to anyone. |
| ... Maybe it was from a gynecological experiment gone wrong.\<br /\>\<br/\>The film is great. It's a great film and I highly recommend it to anyone. It's a great film and I highly recommend it to anyone. |
| ... feeling and atmosphere perfectly, helped in part with some incredible archival footage. Tony Alvais a great film, it is a great film, I highly recommend it to anyone. |

## D   Details of Human Evaluation on HH Dataset

A total of 330 comparison pairs were sampled for RRHF evaluation, involving comparisons between RRHF and good responses (110 pairs), RRHF and PPO (110 pairs), and RRHF and RRHF$_{\text{IP-2}}$ (110 pairs). Out of these, 30 pairs were used to calculate agreement, while the remaining 300 pairs were used for reporting scores. Each crowd-sourced worker was tasked with labeling 130 pairs, consisting of 100 random pairs and 30 common pairs. The average consistency between each pair of reviewers was calculated, revealing that they provided the same annotations for 57.7% of pairs and their annotations did not contradict each other for 84.4% of pairs.

## E   Wombat: Learn from ChatGPT comparison

**Sampling Policy**   We use training data from Alpaca as sampling queries. We sample five different responses for training: $\rho_1, \rho_2$ are generated by ChatGPT, $\rho_3$ is generated by `text-davinci-003`, $\rho_4$ is generated by LLaMA and $\rho_5$ is generated by Alpaca.

**Reward Prompt**   We require ChatGPT to read five responses and score by four different dimensions: Relevance, Correctness, Coherence, and Safety from 1 to 5. We parse the outputs of ChatGPT and calculate the sum as the response reward score. The used prompt is:

*Score different responses separately without explanation and without copying any input from these respects, please scores start from response 1: Relevance (does it relevant to user's query), Correctness (does it contain correct knowledge), Coherence (does it generate fluently and without grammar problems), Safety (does it refuse to answer sex or criminal queries) and give a score for each respect 1-5. Query: ... Response 1: ... Response 2: ... Response 3: ... Response 4: ... Response 5: ...*

**Results**   For 52k training data from Alpaca, we successfully parsed 46k data with reward scores. We use Alpaca as our initial checkpoint and train it by RRHF. We name this model Wombat. The total training time of Wombat is only 4 hours on 8 A100 GPUs. Here we give generated examples of Wombat in Appendix C to show its ability. We can find it has general world knowledge, the ability to answer complex machine learning problems, and the ability to follow instructions.

## F   Example responses generated by Wombat

**Table 12:** Example responses generated by Wombat.

| |
|---|
| **Query:** Give a introduction of wombat |
| **Response:** Wombat is a marsupial found in Australia, Tasmania, and New Zealand. They have a distinctive appearance with a short, stocky body, a broad head, and a short tail. Wombats are herbivores and primarily graze on grasses, leaves, and bark. They are known for their burrowing habits and can dig burrows up to 18 meters deep and 10 meters long. Wombats are a protected species and are considered vulnerable due to habitat loss and predation by introduced species. |
| **Query:** How to design an algorithm to rank responses of language model outputs with aligned human preferences? |
| **Response:** One possible algorithm to rank responses of language model outputs with aligned human preferences is:
1. Collect data: Collect data on human preferences for a given task, such as rating the importance of different features or characteristics of a product or service.
2. Preprocess data: Preprocess the data by removing outliers, missing values, and irrelevant information.
3. Train a language model: Train a language model using the collected data to generate responses based on the preferences of humans.
4. Evaluate model performance: Evaluate the performance of the language model by comparing its responses to the collected data on human preferences.
5. Design a ranking algorithm: Design a ranking algorithm that takes into account the preferences of humans as well as the performance of the language model.
6. Apply the ranking algorithm: Apply the ranking algorithm to the language model's responses to rank them based on their alignment with human preferences.
7. Evaluate the results: Evaluate the results of the ranking algorithm to determine its effectiveness in aligning language model outputs with human preferences.
8. Improve the algorithm: Continuously improve the algorithm by incorporating feedback and refining the preferences data and the ranking criteria. |
| **Query:** Assume you are a teacher, you are teaching a 6-year-old child. Do not give the children direct answers, but try to teach him. Children: How to calculate 6+7? |
| **Response:** Great! Let's count up to 13 together. 6+7 equals 13.
Starting with 6, we can count up by 1 each time.
1, 2, 3, 4, 5, 6, 7, 8, 9, 10, 11, 12, 13.
We've counted up to 13!
So, the answer is 13. |

