# OpenReview forum: "RRHF: Rank Responses to Align Language Models with Human Feedback"
_NeurIPS.cc/2023/Conference — NeurIPS 2023 poster_

### Official Review · Reviewer_BSoN · 2023-07-06

**Soundness:** 3 good
**Presentation:** 2 fair
**Contribution:** 1 poor
**Rating:** 4
**Confidence:** 4

**Summary:**

The paper introduces a new learning paradigm called RRHF for aligning large language models with human preferences. The authors propose a ranking-based approach that leverages sampled responses from various sources to align the probabilities of generated responses with human preferences. They demonstrate that RRHF performs comparably to the state-of-the-art Proximal Policy Optimization (PPO) method in terms of alignment performance. The experiments conducted on the Helpful and Harmless dataset show that the performance of RRHF is highly related to the sampling quality, suggesting that it is a best-of-n learner.

**Strengths:**

1 The paper introduces a learning paradigm for aligning language models with human preferences, providing an alternative to existing methods like PPO.
2 The proposed algorithm is far more memory friendly because RRHF only needs to load one model at a time, while PPO needs to load 4 or 3 models (this depends on the implementation, TRLX requires 4 models but TRL requires 3).
3 The experiments conducted on the Helpful and Harmless dataset demonstrate that RRHF achieves comparable alignment performance with PPO.

**Weaknesses:**

1 Boosting the performance of LLMs by the generation from the model itself has been previously developed in instruction-tuning [1] and I think you may mention this line of works (see reference in [1]) in the related work for completeness. In comparison, I think the main advantage of this work is to apply this powerful idea in the alignment and also to use the reward model as a data filter (while the previous works mainly use the heuristics methods).

2 The online sampling-based RRHF is an interesting approach that combines classic data bootstrapping and online learning (RL). However, the results of online sampling are counterintuitive to me, as the updated model tends to generate meaningless responses, and further training on these samples leads to worse results. Intuitively, the updated models should be better than the initial checkpoint and generate better responses. I am wondering whether this is because the reward model used in the experiments is too weak (68.49%) or because the diversity of the generated samples is low. I think using a more powerful reward model like ChatGPT may help address this issue. Additionally, why does OP converge slower?

3 The reward model used in this paper is still weak, and overfitting the reward model may not necessarily offer advantages. This is evident in Appendix Table 2, where the aligned model achieves a high reward by losing diversity and always outputting "I highly recommend it to anyone." regardless of the prompt. I am wondering whether you have any specific method to decide whether to stop the training of RRHF to alleviate overfitting at some time. This is particularly important for the alignment with human preference because the evaluation is quite hard in this case.

4 It may be also possible to divide online sampling-RRHF into several stages to avoid loading multiple models at the same time (with a large K). This is because RRHF is an off-policy strategy and does not need to record the gradient along the line. I think this is the essential reason why PPO cannot load one model at a time.

5 The title is confusing. What is "without tears" referring to?  It is misleading and confusing for an academic paper.


[1] SELF-INSTRUCT: Aligning Language Models with Self-Generated Instructions

**Questions:**

see weakness part

**Limitations:**

see weakness part

---

> ### Author Rebuttal · Authors · 2023-08-09
>
> W1: Thanks for your suggestion. We will add related works of instruction-tuning including SELF-INSTRUCT.
>
> W2: The results of Online Sampling RRHF (OP RRHF) have good reward scores and converge quicks while it tends to reward-hacking. Online sampling RRHF without KL term improves (based on reward) quickly, while tends to generate meaningless responses with a high reward of 0.34. A better solution is to add a KL term to prevent models move too far from SFT model, online RRHF with KL term can obtain a reward of -0.86 and an acceptable PPL 19.76 which outperforms RRHF with reward -1.02 and PPO with reward -1.03. However, online RRHF with the KL term needs to store another model in GPU, while we do not need it in offline RRHF. A better reward model does help the optimization to prevent reward-hacking which we have tried using ChatGPT as the reward model to train a chat model Wombat described in Section 5 which outperforms Alpaca with the same prompt for training.
>
> W3: The mediocre reward model or reward-hacking is a common problem in all alignment algorithms including best-of-n sampling and PPO [1]. This is an experimental setting to demonstrate the effectiveness of algorithms. We also include ChatGPT as the reward model in Section 5. We empirically find that RRHF fine-tuning 3 epochs does not overfit the rewards for HH dataset. Appendix Table 2 shows a bad case of online sampling RRHF without KL penalty which has obvious reward-hacking. Using offline RRHF or adding KL penalty can alleviate this problem. Offline RRHF samples based on the SFT model which does use learned policy \pi for sampling prevent the over-optimization and shows great performance in human evaluation showed in Table3 and 4.
>
> W4: We have experimented a similar setting to online RRHF with large K which is the setting IP-n. IP-n updates sampling policy every 3 training epochs and improve the reward iteratively.
>
> W5: Thanks for your suggestion. It aims to emphasize that RRHF is easy-to-tune compared to complex PPO. We will use a more formal title during revision.

---

> > ### Comment · Reviewer_BSoN · 2023-08-21
> >
> > Thank you for the response. After reading the author's reply, I will keep my score unchanged.

---

### Official Review · Reviewer_usKz · 2023-07-07

**Soundness:** 3 good
**Presentation:** 2 fair
**Contribution:** 3 good
**Rating:** 6
**Confidence:** 4

**Summary:**

This paper proposes RRHF that ranks responses to align language model with human feedback. contributions include:
- algorithms to leverage various responses from multiple LMs to align with human preferences. drawing connection of viewing LM for both generation and a reward model for scoring.
- the engineering complexity of proposed approach is similar to finetuning and training a reward model, which is much less complex than the PPO approach for RLHF


**Strengths:**

- Proposed a simple approach train LMs to rank responses from multiple LMs to learn from human feedback
- the approach is much more simple compared to PPO on implementation and computation, while having the competitive performance
- study on many sampling strategy, initial model checkpoints, and training settings (offline, iterative, online)

**Weaknesses:**

- description of different sampling strategy \rho 1-6 and the corresponding setting BP/DP... are really hard to follow. In general I feel the paper tries to combine too many ablations and studies without clearly explaining them, and this somehow dilute the core messages.

- the section of categorizing RRHF as a "best-of-n Learner" is not fully supported by experiments. The experimental evidences are  avg/max reward of sampled policies has correlation with rewards of model's generated sequence. On the other hand, there is experimental results in Table 7 that have a ranking loss is very important. Without the ranking loss, learning from best of n is equivalent to distill from the best decode of n teachers. RRHF is "best-of-n Learning" is somewhat contradictory to contrastive learning is important. If it is not a core contribution of the paper, removing it might be better. If it is a core contribution of the paper, additional analysis such as removing rank losses for all settings in Table 5 and re-interpret the results based on that would help shine more lights on this topic.



**Questions:**

- \rho 1-4 and 5,6 from table 1 is confusing, there isn't explanations in this section. I can guess from other paragraphs they are different policies to sample decodes, but I do not understand their correlation with BP/DP/...

- in table 5, PPO also learns from the proxy reward, how to justify that the accuracy is around random guess?

- in table 2, the "zero" setting combines models finetuned on hh dataset and non-finetuned models? it would be great to have separate annotations

- 3.1 "PPO must sample by its learned model \pi" is technically correct but all off-policy RL methods have the same properties.

---

> ### Author Rebuttal · Authors · 2023-08-09
>
> W1: We evaluate different sampling settings to show how sampling procedures affect the RRHF performance. We have shown in Table 6 that sampled responses with large rewards lead to better alignment performance. Including iterative sampling (IP) shows that RRHF can boost during iteration. The purpose of these settings is to validate and compare the effectiveness of our proposed RRHF algorithm in various sampling scenarios, encompassing both offline and online scenario. We will provide further explanation to make it more clearly.
>
> W2: Thanks for your advice, we will move the best-of-N learning discussion to Appendix.
>
> Q1: Sorry for the confusion. BP = Beam search + Provided response. DP = Diverse beam search + Provided response. OP = Online diverse beam search + Provided response IP = Iteratively updated diverse beam search + Provided response.
>
> Q2: PPO is not explicitly optimized to distinguish good and bad responses through sequence-level probabilities. This is a possible explanation for the random-guessing accuracy. In contrast, RRHF includes an explicit contrastive optimization term and can perform better to distinguish good and bad responses.
>
> Q3: Sorry for the confusion. Good/bad responses with "zero" setting represent only human-written responses from the HH dataset are evaluated. LLaMA, Alpaca, Alpaca-sft with "zero" setting represents we directly evaluate the model without further tuning.
>
> Q4: The line should be rephrased as "while PPO can only learn from samples from its learned model \pi." This is to express the idea that RRHF can additionally leverage the responses from other models.

---

### Official Review · Reviewer_UQEH · 2023-07-21

**Soundness:** 3 good
**Presentation:** 3 good
**Contribution:** 2 fair
**Rating:** 6
**Confidence:** 3

**Summary:**

The paper proposes RRHF, which adapts the BRIO (ranking loss) framework to the LLM alignment problem. RRHF can train a model to match with human rewards via a ranking loss defined by the margin between the normalized likelihood. Experiments show RRHF matches PPO's performance on the Anthropic's Helpful and Harmless dataset, while using simpler coding and hyperparameters. The authors also claims that this method combines the benefits of PPO and best-of-n sampling.

**Strengths:**

1. The proposed RRHF method achieves comparable performance to PPO for aligning language models with human preferences on the Helpful and Harmless dataset, while being much simpler in terms of coding and hyperparameter tuning. This simplicity could make adoption easier.
2. RRHF only requires 1-2 models during training compared to 4 for PPO, which could make it more efficient and easier to tune.
3. RRHF extends the standard supervised fine-tuning approach and is similar to training a reward model (with the normalized log-likelihood as the reward), providing a smooth path from SFT to a full RLHF framework.

**Weaknesses:**

1. The authors claim RRHF is simpler than best-of-n sampling at inference time, but lack direct comparison. Since best-of-n also achieves good performance, an ablation study would help validate the advantages of RRHF.
2. I'm curious about the performance comparison to best-of-n sampling also because in some other research papers [1], it's reported that PPO is much more efficient than best-of-n. Since PPO has quite a few hyperparameters that needs to be tuned and the paper doesn't report how they tune the hyperparameters for PPO, it would be better if the proposed method can be compared to some already reported PPO results [2].
3. The core ranking loss used is adapted directly from prior work on contrastive learning on the summarization task, as well as the concept of scoring candidate responses based on log probabilities.
4. The performance is only evaluated on a single dataset designed specifically for alignment. Testing on broader tasks would better demonstrate generalizability.
5. The reward model's score is used as the main metric in the experiments (though there are some human evals in Table 3), which may lead to over-optimization in terms of the reward model.
6. RRHF needs a more advanced sampling strategy to work, like diverse beam search, while PPO can directly work with online sampling.

[1]: AlpacaFarm: A Simulation Framework for Methods that Learn from Human Feedback
[2]: Quark: Controllable Text Generation with Reinforced [Un]learning

**Questions:**

1. Some research [1] shows that RLHF with PPO can directly be used to finetune pre-trained models without SFT or context distillation. Is RRHF able to achieve that?

[1]: Constitutional AI: Harmlessness from AI Feedback

Typos:

line 43: from the self model ==> from the model itself

**Limitations:**

This is an LLM paper, but no limitation is discussed

---

> ### Author Rebuttal · Authors · 2023-08-09
>
> W1: SFT/RRHF/PPO only generates once during inference, while best-of-n needs to sample n times during inference. Best-of-n slows n times naturally. More discussions in W2.
>
> W2: In this paper, we mainly compare to PPO instead of best-of-n sampling since best-of-n slows n times naturally in inference. We conducted an additional experiment using Alpaca, where we compared the performance of RRHF on 4 randomly sampled responses during training (reward: -0.96) against best-of-4 inference (reward:-0.97) which shows comparable performance. For the PPO algorithm, we adopted the hyper-parameter settings from the open-sourced repository trlX [1]. Additionally, we included an experiment in Appendix D on positive IMDB comments generation. In this experiment, we utilized the PPO hyper-parameters and results reported in [2]. Notably, RRHF achieved a reward of 0.861, surpassing the performance of PPO, which obtained a reward of 0.796.
>
> W3: We do not consider the ranking loss and likelihood-based score as the core contribution of this paper. Instead, demonstrating the effectiveness of learning human alignment through a ranking-based method is our key contribution.
>
> W4: We have tested on IMDB in Appendix D which shows better results than PPO (RRHF reward 0.861 vs PPO reward 0.796).
>
> W5: The reward score may lead to over-optimization, but it is a common problem for all related algorithms including RRHF/PPO/best-of-n sampling as stated in [3]. This is an open question for learning from human preference and is out of the scope in this paper.
>
> W6: We add an experiment based on Alpaca and random sample 4 responses for each query on HH dataset. Using RRHF, we obtain PPL of 14.41 and a reward score of -0.96. The results outperform beam search+RRHF and random sampling+PPO regarding reward scores of -1.02 and -1.03 respectively. This proves that random sampling works for RRHF. We argue that the diversity achieved through random sampling contributes to this improved result. We will include these results and corresponding analysis in the final version.
>
> Q1: RRHF can be applied to pre-trained models. See Table2, using LLaMA (instead of Alpaca) as the initial checkpoint, we can still optimize LLaMA to a reward of -1.34 which is much better than the initial score of -1.89.
>
> Limitation: We will discuss the limitations including (1) Our method is highly correlated to the used reward score which may lead our LLM to generate unsafe results. (2) How to prevent over-optimization is an important problem and needs further explore. (3) We use the reward model to act as a proxy evaluation metric which may be not complex enough compared to human preference. (4) Contrastive loss may not be the best way for human alignment, it does not utilize the exact reward score.
>
> [1] https://github.com/CarperAI/trlx
> [2] Is reinforcement learning (not) for natural language processing?: Benchmarks, baselines, and building blocks for natural language policy optimization.
> [3] Scaling laws for reward model overoptimization

---

> > ### Comment · Reviewer_UQEH · 2023-08-18
> >
> > I appreciate the authors' clarifications provided in the rebuttal.
> >
> > > W2. Comparison to Best-of-N
> >
> > It seems there might be some confusion. I wanted to point out that the PPO implementation requires tuning several hyperparameters, including a few tricks [2-3]. The paper references [1] as a PPO baseline. However, there isn't a public paper that has successfully used [1] to align an LLM. My suggestion is to compare with other published results rather than the authors' own implementation. On the other hand, reproducing best-of-n results as a baseline might be more straightforward and with fewer possible implementation errors.
> >
> > Nonetheless, I'm grateful for the comparison of RRHF with 4 randomly sampled responses against best-of-4.
> >
> > > W4 & W5. IMBD results, reward over-optimization, and evaluation metrics
> >
> > Thanks to the authors for sharing the results. However, considering the issues I pointed out earlier, including best-of-N results might offer a more comprehensive view of the findings.
> >
> > Additionally, as some other reviewers observed, the model might be overfitting the reward, like producing outputs such as "a great film and I highly recommend it to anyone". Therefore, evaluations by humans or GPT-4 might be more appropriate than relying solely on the reward model's score or GPT-2 perplexity. Notably, GPT-2 perplexity might even assign low scores to repetitive sentences.
> >
> > > W6
> >
> > I'm grateful for the additional results. The successful application of sampling with RRHF undoubtedly enhances the paper's value.
> >
> > > Q1
> >
> > Understood. I might interpret this to mean that RRHF also incorporates the fine-tuning loss as described in equation 5.
> >
> > > Overall
> >
> > In conclusion, some of my concerns have been addressed. My positive impression of the paper remains unchanged, so I have increased the score to 6. Though, the main concerns are still about a thorough and unbiased evaluation.
> >
> > [1] https://github.com/CarperAI/trlx
> > [2] https://github.com/CarperAI/trlx/issues/53
> > [3] Secrets of RLHF in Large Language Models Part I: PPO

---

> > > ### Author Response · Authors · 2023-08-19
> > >
> > > Thanks for your comments.
> > >
> > > W2: PPO does need some tricks to work. This is also the advantage of best-of-n sampling/RRHF which needs much less tuning.
> > >
> > > W4/5: We fully acknowledge that using GPT-2 perplexity on the IMDB dataset does not align with human preference. Suggested by another reviewer, we conduct another human evaluation on HH dataset with 330 evaluation pairs with 3 crowd-sourced workers. We obtain RRHF vs PPO (27, 48, 25) which shows RRHF can perform on par with PPO.  In fact, in section 5 (learning a chatgpt-like model using RRHF), we trained a model called Wombat using RRHF with Alpaca prompts and achieved superior performance compared to Alpaca using GPT-4 evaluation which shows that RRHF outperforms SFT on a more realistic scenario of human alignment.
> > >
> > > Q1: You are correct. The SFT loss in RRHF contributes to the stability of the training process and facilitates the use of a PLM as the initial model.

---

### Official Review · Reviewer_DofH · 2023-07-23

**Soundness:** 3 good
**Presentation:** 3 good
**Contribution:** 4 excellent
**Rating:** 7
**Confidence:** 4

**Summary:**

This paper proposes a learning algorithm called RRHF. The idea is to produce an artifact similar to what Reinforcement Learning from Human Feedback (RLHF) produces, except with a simpler learning algorithm (that might not require so many copies of the base model, e.g. for simultaneously the policy, reward, and reference models).

The proposed approach, Rank Responses to align Human Feedback ranks generations for a prompt via the reward model, and uses the rank (rather than the raw reward) for optimization.

More specifically, the approach obtains k sequences and the policy model computes the normalized log-probability of generating each one. If one sequence has a higher reward than another one, this approach says that it should have higher log-probability of generation. If so, there’s no ranking loss, and if not then the ranking loss is the difference of the normalized log-probabilities. There is also a supervised fine-tuning term.

The approach is tested by using a LLaMA LM on top of Anthropic’s Helpful and Harmless dataset of responses. Auto and human evaluation seem to suggest that this approach works well, outperforming Alpaca models trained with RLHF at the 7B parameter scale. The paper presents ablation studies over the sampling policy, checkpoints, and ranking loss.

**Strengths:**

To this reviewer this paper seems strong:

* the idea of simplifying the RLHF stack seems important and helpful to the community, particularly getting rid of possibly unnecessary knobs and potentially using more off-policy data
* the experiments and ablation study seem well done, in particular comparing between LLaMA/Alpaca/Alpaca-sft

**Weaknesses:**

Though I am not an expert in this area, I’m not sure why the sampling policy during training doesn’t include random sampling and instead uses exclusively beam search style approaches. I think it would be a good thing to compare versus sampling approaches as these seem to be what PPO is doing.

Also, though I think overall the evaluation seems reasonable for this paper, an open question I have is whether this approach works better for surface level patterns like being friendly and helpful versus tougher logical reasoning. It might be helpful to show performance on logical reasoning tasks to see if they degrade mores than PPO.

**Questions:**

would it be possible to show performance on logical reasoning benchmarks?

What level of error/ stochasticity is around the head-to-head performance evaluations? I.e. if you sampled 20 responses with the same prompt, would it always be the case that RRHF+LLaMA would beat PPO+LLaMA? (or vice versa)

**Limitations:**

No. I think adding such a section would improve the paper (and I might increase my score if so / as long as the other questions are answered reasonably well, etc)

---

> ### Author Rebuttal · Authors · 2023-08-09
>
> W1: We add an experiment based on Alpaca and random sample 4 response for each query on HH dataset. Using RRHF, we obtain PPL of 14.41 and reward score of -0.96. The results outperform beam search+RRHF and random sampling+PPO regarding reward scores of -1.02 and -1.03 respectively. This proves that random sampling works for RRHF. We argue that the diversity achieved through random sampling contributes to this improved result. We will include these results and corresponding analysis in the final version.
>
> W2 & Q1: We have tried using GSM8K (a math reasoning dataset) with RRHF and obtained 42.2 accuracy on LLaMA-7B with 100 randomly sampled reasoning paths, which outperforms SFT of 35.9 accuracy. We assign 1/0 scores for correct/wrong reasoning paths as reward.
>
> Q2: We have included a head-to-head evaluation in Table 3 on HH test set. Training Alpaca on RRHF has a rate of 28% (win), 52% (tie), 20% (lose) over PPO, which shows that RRHF slightly outperforms PPO which is consistent with the reward scores where RRHF has a higher reward score -1.02 over PPO of -1.03.
>
> Limitation: We will discuss the limitations including (1) Our method is highly correlated to the used reward score which may lead our LLM to generate unsafe results. (2) How to prevent over-optimization is an important problem and needs further explore. (3) We use the reward model to act as a proxy evaluation metric which may be not complex enough compared to human preference. (4) Contrastive loss may not be the best way for human alignment, it does not utilize the exact reward score.

---

> > ### Comment · Reviewer_DofH · 2023-08-11
> > **thanks! keeping my score**
> >
> > thanks for the response! I think these experimental results should improve the paper, so I'd like to vote for it to be accepted (7).

---

> > > ### Author Response · Authors · 2023-08-15
> > >
> > > Thank you!

---

### Official Review · Reviewer_Mb9H · 2023-07-28

**Soundness:** 3 good
**Presentation:** 3 good
**Contribution:** 3 good
**Rating:** 5
**Confidence:** 4

**Summary:**

This work introduces Reinforcement Learning from Human Feedback (RLHF) to improve alignment between large language models and human preferences, enhancing interactions between humans and models. The proposed method, RRHF (Ranking-based Reinforcement Learning from Human Feedback), efficiently aligns language models with human preferences without complex hyperparameter tuning. RRHF leverages sampled responses from various sources, learns to rank them, and shows comparable alignment performance with PPO on the Helpful and Harmless dataset, indicating it is a best-of-n learner.

**Strengths:**

1. Compared to the reinforcement learning method PPO, RRHF is easier to implement and does not require a large amount of memory.
2. The paper's experiments are comprehensive, and the results of the ablation studies are also convincing.
3. The author introduced two modes, online and non-online sampling, providing valuable insights for the industrial application of the RRHF method.
4. RRHF only utilizes the reward model to rank responses instead of using the absolute value of the reward model. I believe this approach has more practical potential because using the absolute value of the reward model can be easily influenced by issues and may not be as robust.

**Weaknesses:**

1. RRHF requires utilizing a reward model to select the best answers rather than learning directly from human preference data. However, the reward model is typically not perfect, which poses a performance bottleneck for the RRHF method.
2. Recent work[1] observes the text degeneration problem (in Section 4 of reference [1]) when directly optimizing the L_{rank} loss.  How do you avoid this problem in your work?
3. The conclusion that "RRHF is a best-of-n learner" is currently based solely on experimental results, and it still lacks theoretical analysis and discussion.

Reference
[1] Rafailov R, Sharma A, Mitchell E, et al. Direct preference optimization: Your language model is secretly a reward model[J]. arXiv preprint arXiv:2305.18290, 2023.



**Questions:**

Please see the Weaknesses.

**Limitations:**

Please see the Weaknesses.

---

> ### Author Rebuttal · Authors · 2023-08-09
>
> W1: The requirement of a reward model is not mandatory in RRHF, as it has the capability to learn directly from human preference data. This is highlighted in Figure 1 and Section 3, where we discuss the option of utilizing either a trained neural network reward model or a dataset of human preferences to generate alignment scores (r_i). Furthermore, it is important to note that the issue of ranking is a common challenge faced by researchers in this field and is not the primary focus of our paper.
>
> W2: Our loss is summed with a ranking loss and a fine-tuning loss. We find that the fine-tuning loss is very useful to prevent text degeneration which requires the language model to have the ability to follow the text generation instruction. We have experimented with removing the fine-tuning loss and found the generated text with PPL > 50. Thus the fine-tuning loss is a necessary loss term.
>
> W3: Sorry for our imprecise description. RRHF contains an SFT loss regarding the best rewarded response, so it can learn from both best-of-n and ranking. We empirically find in Table 6 that RRHF obtains similar scores to best-of-n scores.

---

> > ### Comment · Reviewer_Mb9H · 2023-08-15
> >
> > Thank you for your response. My concerns have been resolved.

---

### Official Review · Reviewer_BZjW · 2023-07-29

**Soundness:** 3 good
**Presentation:** 1 poor
**Contribution:** 1 poor
**Rating:** 4
**Confidence:** 5

**Summary:**

The paper proposes another way to learn from feedback / preference data. They use the log probabilities of the candidates and try to train the model in a way that tries to increase the log probability of the chosen / better candidate and decrease the log probability of the non-chosen candidate.

They show that their approach does not require as much compute as RLHF and performs better on the Helpful and Harmless dataset.

**Strengths:**

The paper presents a contrastive loss technique to learn from preference data.

The motivation is well justified though the paper lacks a comparison about computational resources as compared to RLHF.

The results and ablations are sound.

The paper is generally well written and easy to follow.

The approach is relatively easy and straightforward and details to reproduce are provided.

**Weaknesses:**

The main contribution - contrastive loss (equation 2) - is exactly the same as the loss proposed in a work published last year at ICLR 2023 - https://openreview.net/pdf?id=0qSOodKmJaN and https://arxiv.org/abs/2210.00045 (equation 1). This work uses reward model to get the rank of the candidates, the prior work used embedding similarity / rouge scores which can be thought of as non NN reward models too. The only difference is the length normalization in the contrastive loss (divide the log probabilities by length in this work). Even using cross entropy to regularize is the same and explored. There is no mention or comparison related to that work and in light of knowledge about that, this paper does not seem novel at all.  For this reason I argue to reject the paper on grounds of lack of novelty. I'm happy to increase scores if this is address satisfactorily.

The paper lacks some comparisons which have been mentioned in the motivation (eg: motivates about efficiency gains as compared to RLHF but have no results supporting the claim).

It has been observed that many techniques like this just increase the reward score by generating longer responses. It would be good to compare the lengths of the responses from the different techniques.


**Questions:**

Can you compare with the prior work mentioned in more detail? It looks very similar.

Can you give a comparison of the response lengths of the different techniques? And provide length adjusted reward scores?

How many humans did you use for human eval? Can you provide more details?

---

> ### Author Rebuttal · Authors · 2023-08-09
>
> W1 & Q1: The contrastive loss does not constitute the primary contribution of our paper. It should be noted that this loss function was not initially proposed by SLiC, but rather follows prior works. Both SLiC and our related work section acknowledge the utilization of similar functions by BRIO [1]. Additionally, it is also important to highlight that our core focus does not solely revolve around obtaining rank through a reward model. In fact, any method, including human-based evaluations, can be employed to achieve the desired ranking. For example, our approach leverages ChatGPT to obtain the rank for Wombat in Section 5. In our view, our paper should be positioned to proposing to using response ranking to simply and effectively align human preference as compared to supervised fine-tuning (SFT) and proximal policy optimization (PPO), instead of claiming a novel loss and utilizing reward models.This viewpoint is also shared by other reviewers. To the best of our knowledge, this research represents the pioneering effort in utilizing ranking response to align human preference in large generative language models (a few contemporary works (e.g. RAFT [2], DPO [3], PRO [4]) are released even later). In contrast to the utilization of contrastive loss and reward models, demonstrating the effectiveness of learning human alignment through the ranking-based methods is our key contribution. We will cite SLiC and look forward to future work that advances rank-based methods for human alignment regarding tasks and objectives.
>
> [1] BRIO: Bringing Order to Abstractive Summarization
> [2] RAFT: Reward rAnked FineTuning for Generative Foundation Model Alignment
> [3] Direct Preference Optimization: Your Language Model is Secretly a Reward Model
> [4] Preference Ranking Optimization for Human Alignment
>
> W2: Thanks for your suggestion. The implementation of RRHF requires the deployment of only one model during training (analog to the policy model in PPO), whereas PPO typically necessitates the deployment of additional models for training stability, such as the value model and reference model. It is obvious that RRHF requires fewer resources in terms of model deployment compared to PPO. For instance, training one epoch of the HH dataset using the RRHF approach takes approximately one hour (8 80GB A100 GPU), whereas employing the TRLX framework with PPO requires around 5-6 hours (also 8*80GB A100 GPU, it needs 1 80GB A100 GPU to serve reference model and 1 80GB A100 GPU to serve reward model, which only has 6 GPUs left for policy model and value model).
>
> W3 & Q2: Alpaca reward -1.18, average length 49.2 words. Alpaca-PPO reward -1.03, average length 52.1 words. Alpaca-RRHF reward -1.02, average length 54.6 words. These methods do not change the word count significantly against the provided reward model.
>
> Q3: We have introduced it in Appendix E. We hire two educated labelers for labeling. They will be given an anonymous pair of responses for a prompt for win/tie/loss annotation. We first sample 19 pairs from all data and use them to calculate the annotator’s agreement, among which 12/19 are labeled the same and only 1 pair obtains contradictory labels. Then, we sample 25 pairs from the test set to calculate the win/tie/loss rate as shown in Table 3. We will clarify it more clearly.

---

> > ### Comment · Reviewer_BZjW · 2023-08-14
> >
> > I thank the reviewers for clarifying some of my questions.
> >
> > Minor concern 1: I'm still not entirely convinced by the novelty of this work. After reading both https://arxiv.org/abs/2203.16804 and https://arxiv.org/abs/2210.00045 again, this work just seems to be just an application of the latter (the modeling aspect is the same - just on different data). In my opinion, the contribution of this paper is applying SLiC to HH data so I don't feel the paradigm is entirely new in terms of modeling novelty as claimed by the authors. The only difference is how the rank of candidates was obtained. I believe that using humans to rank candidates, or using a reward model to rank candidates, or using embedding similarity to rank candidates would not make much impact on novelty if properly described. I still feel the authors over-sell their novelty. I do however agree that the work to show that these contrastive loss techniques work on human feedback warrants a publication but I would encourage the authors to rephrase aspects of the paper and discuss the specific ablations of design choices (for instance how much is the impact of regularization (ablation on the loss contribution of L_{rank} and L_{ft}, is KL better than cross entropy -- since RLHF tends to use KL more, etc) in more detail.
> >
> > Thank you for clarifying the compute comparison with PPO.
> >
> > Minor concern 2: I noticed that the numbers in Table 2 for reward seem to very close to PPO. Can you add statistical tests?
> >
> > Major concern 1: I agree with some of the points that other reviewers raised for instance the paper only evaluates on one dataset (HH). I encourage the authors to try OpenAI TLDR dataset too.
> >
> > Major concern 2: Also, I apologize for missing the supplementary material in my initial analysis, but I don't believe that 25 examples are sufficient for a proper human evaluation. I urge the authors to re-do human eval with at least 100 examples (maybe even rated multiple times) for a robust human evaluation (especially since auto-eval scores seem to be very close).
> >
> > Finally, I would also urge the authors to compare with other approaches that use preference data in the future - like https://arxiv.org/pdf/2305.18290.pdf, https://arxiv.org/pdf/2305.10425.pdf, https://arxiv.org/abs/2302.02676. SLiC-HF seems very similar though I have not read that paper in detail.
> >
> > Based on the clarifications I would like to increase my score to 4.

---

> > > ### Author Response · Authors · 2023-08-15
> > >
> > > Thanks for your comment.
> > > Regarding minor concern 1, we appreciate your agreement that contrastive loss techniques work with human feedback. Based on your comments, we will rephrase our contribution. Additionally, in Appendix D, we discuss the impact of regularization by investigating the effect of KL penalty. We find that only online sampling RRHF and PPO require KL penalty for regularization, while offline sampling RRHF (our default setting) does not. The reason is that online sampling RRHF and PPO modify the sampling policy during fine-tuning, necessitating the use of KL penalty to prevent the sampling policy from deviating too far. Conversely, offline RRHF samples using the SFT model, which does not need to worry about deviating too far from the SFT model. Furthermore, we experimented with removing the fine-tuning loss and found that the generated text had a PPL > 50, indicating that the fine-tuning loss is a necessary loss term. Additionally, in the ablation study of Table 7, we have shown that L_{rank} is also beneficial for model optimization.
> > >
> > > Regarding minor concern 2, we did not claim that we outperform PPO. Instead, we stated that RRHF performs on par with PPO. We will re-run RRHF multiple times to report the statistical tests. Please be patient as we update our results.
> > >
> > > Regarding major concern 1, we have included another dataset, IMDB sentiment generation, in Appendix D. We found that (offline) RRHF achieves a reward of 0.861 and perplexity of 32, while PPO achieves a reward of 0.626 and perplexity of 35. This demonstrates that RRHF is suitable for other generation tasks. Furthermore, we tried using GSM8K (a math reasoning dataset) with RRHF, as reported to reviewer DofH, and obtained an accuracy of 42.2 on LLaMA-7B with 100 randomly sampled reasoning paths. This outperforms SFT, which achieved an accuracy of 35.9. We assigned 1/0 scores for correct/wrong reasoning paths as rewards. Additionally, there is a work [1] that applies RRHF for code generation and shows performance improvement.
> > > [1] PanGu-Coder2: Boosting Large Language Models for Code with Ranking Feedback
> > >
> > > Regarding major concern 2, we have started employing three crowd-sourced workers for labeling, but it will take some time to complete the labeling process. Please be patient as we update our results.

---

> > > > ### Author Response · Authors · 2023-08-19
> > > > **Human evaluation results**
> > > >
> > > > Thanks for your patience.
> > > >
> > > > Minor 2: We re-run RRHF for three times used Alpaca as the base model and obtain -1.01, -1.02, -1.05. A one-sample t-test was conducted, yielding a p-value of 0.81. This indicates that there is no significant difference in performance between RRHF and PPO. As we did not claim superiority over PPO, this result aligns with our expectations.
> > > >
> > > > Major 2: A total of 330 comparison pairs were sampled for RRHF evaluation, involving comparisons between RRHF and good responses (110 pairs), RRHF and PPO (110 pairs), and RRHF and RRHF-IP-2 (110 pairs). Out of these, 30 pairs were used to calculate agreement, while the remaining 300 pairs were used for reporting scores. Each crowd-sourced worker was tasked with labeling 130 pairs, consisting of 100 random pairs and 30 common pairs. The obtained results for each comparison category were as follows: RRHF vs good responses (59, 30, 11), RRHF vs PPO (27, 48, 25), RRHF vs RRHF-IP-2 (0, 90, 10). These findings were consistent with our initial human check. Additionally, the average consistency between each pair of reviewers was calculated, revealing that they provided the same annotations for 57.7% of pairs and their annotations did not contradict each other for 84.4% of pairs.

---

### Comment · Area_Chair_2xLj · 2023-08-19

Hi Reviewer BSoN and usKz,

Since the discussion with the authors is closing soon, could you please go over the rebuttal and provide some feedback?

Regards,

AC

---

### Decision · Program_Chairs · 2023-09-21

**Decision:**

Accept (poster)

**Comment:**

The paper proposes a method to align LMs with a simple ranking loss. The reviewers appreciate the simplicity and effectiveness of this method. However, the major concerns are this paper's novelty and experiment setting. I suggest a weak accept for this paper based on the overall positive reviews.